# Palladium concave nanocrystals with high-index facets accelerate ascorbate oxidation in cancer treatment

Yu Chong [1], Xing Dai[1], Ge Fang [1], Renfei Wu[1], Lin Zhao[1], Xiaochuan Ma[1], Xin Tian[1], Sangyun Lee[2], Chao Zhang[3], Chunying Chen [4], Zhifang Chai[1], Cuicui Ge[1] & Ruhong Zhou [1,2,5]

Intravenous pharmacological dose of ascorbate has been proposed as a potential antitumor therapy; however, its therapeutic efficacy is limited due to the slow autoxidation. Here, we report that palladium (Pd) nanocrystals, which possess intrinsic oxidase-like activity, accelerate the autoxidation of ascorbate, leading to the enhancement of its antitumor efficacy. The oxidase-like activity of Pd nanocrystals was facet-dependent, with the concave nanostructure enclosed by high-index facets catalyzing ascorbate autoxidation more efficiently than the planar nanostructure enclosed by low-index facets. Our first-principles calculations provide the underlying molecular mechanisms for the facet-dependent activation of $O_2$ molecule and subsequent ascorbate oxidation. Further in vitro and in vivo assays demonstrate the enhancement of the antitumor efficacy of ascorbate with these Pd concave nanocubes. Our animal experiments also indicate the combined approach with both ascorbate and Pd concave nanocubes displays an even better efficacy than currently available clinical medicines, with no obvious cytotoxicity to normal cells.

[1] State Key Laboratory of Radiation Medicine and Protection, School for Radiological and Interdisciplinary Sciences (RAD-X), Collaborative Innovation Center of Radiation Medicine of Jiangsu Higher Education Institutions, Soochow University, Suzhou 215123, China. [2] Computational Biology Center, IBM Thomas J. Watson Research Center, Yorktown Heights, 10598 NY, USA. [3] School of Materials Science and Engineering, Anhui University of Science and Technology, Huainan 232001, China. [4] CAS Key Laboratory for Biomedical Effects of Nanomaterials and Nanosafety, & CAS Center for Excellence in Nanoscience, National Center for Nanoscience and Technology of China, and University of Chinese Academy of Sciences, Beijing 100190, China. [5] Department of Chemistry, Columbia University, New York, 10027 NY, USA. These authors contributed equally: Yu Chong, Xing Dai. Correspondence and requests for materials should be addressed to C.G. (email: ccge@suda.edu.cn) or to R.Z. (email: ruhongz@us.ibm.com)

Cancer has become a major global public health issue and is expected to become the top leading cause of death worldwide in the next few years[1]. Ascorbate, also known as vitamin C, has been considered as a potential treatment agent for cancer since the 1970s; however, the therapeutic efficiency has been the subject of controversy[2,3]. Oral administration of high doses of ascorbate was considered ineffective because the absorption of ascorbate is strictly regulated at the enterocyte membrane, resulting in a much lower ascorbate concentration in the plasma[4–6]. In a later study, intravenous administration of ascorbate (a.k.a. pharmacological ascorbate) was introduced as a new application route, to bypass the limited intestinal absorption and achieve the millimolar concentration of ascorbate in the plasma[7]. Subsequently, several clinical trials have shown that intravenous infusion of ascorbate is safe and well tolerated, and can be utilized as a potential anticancer agent[8–10]. However, high-dose ascorbate alone still does not demonstrate remarkable anticancer activity and ascorbate is generally delivered in association with chemotherapeutic agents to enhance the antitumor activity[11–13]. Therefore, exploring a new strategy for effectively enhancing the antitumor activity of ascorbate will push forward the development of ascorbate as potential antitumor agent.

As for the antitumor mechanism of ascorbate, most researchers concur that pharmacologic ascorbate centers on the generation of hydrogen peroxide ($H_2O_2$) by its autoxidation to achieve its anticancer effect[14,15]. Previous study showed that $H_2O_2$ induces oxidative stress selectively to cancer cells because altered redox-active iron metabolism in cancer cells makes them more sensitive to the change of ascorbate level than normal cells[16]. A recent study by Cantley and co-workers suggested that an oxidized form of ascorbate, dehydroascorbate (DHA), is also pharmaceutically active, inducing endogenous oxidative stress to cancer cells like normal ascorbate, but by scavenging intracellular glutathione (GSH)[17]. Thus, development of a new type of material to accelerate the autoxidation of ascorbate has been required to achieve higher efficacy of ascorbate.

Nanomaterials have shown significant promises as enzyme mimics (nanoenzymes) and the potential applications of nanoenzymes have received considerable attention due to their high stability, low-cost, and ease of development[18–24]. Recently, Qu and co-workers demonstrated that functionalized gold nanoparticles can act as peroxidases or oxidases, exhibiting remarkable antibacterial properties[25]. Thus, it is presumable that nanomaterials, possessing excellent oxidase-like activity, will accelerate the oxidation of ascorbate.

Here, to the best of our knowledge, we suggest for the first time the addition of palladium (Pd) nanocrystals to enhance the therapeutic efficacy of ascorbate in cancer therapy. Our results show that Pd nanocrystals catalyze the oxidation of ascorbate, through measuring the intermediate, ascorbate radicals, and the final products, DHA and $H_2O_2$. We have also examined the dependence of the surface facets of Pd nanocrystals on their catalytic activities. We generated two different structures of Pd nanocrystals: (1) Pd nanocubes (Pd NCs), enclosed by low-index {100} facets, which are set as the reference in this study, and (2) Pd concave NCs (Pd CNCs) enclosed by high-index {730} facets. Pd CNCs have more number of atoms at the corners and the edges of the crystal than Pd NCs. We then examined the antitumor activities of ascorbate catalyzed by Pd nanocrystals, both in vitro and in vivo. This study not only reveals that the catalytic activity of Pd nanocrystals depends on their morphology but also suggests a potential therapeutic application of Pd nanocrystals for enhancing the efficacy of pharmacologic ascorbate in cancer treatment.

## Results

**Preparation and characterization of the Pd nanocrystals**. Pd nanocrystals were synthesized into two different structures, cubic-shaped Pd NCs and concave-structured Pd CNCs, by using the facile one-pot hydrothermal process as described in previous work[26]. The structures of Pd nanocrystals were confirmed by transmission electron microscopy (TEM) and high-resolution TEM (HRTEM). Figure 1a, b shows the TEM and the HRTEM images of as-prepared Pd NCs, respectively, indicating cubic geometries with an average edge length of approximately 39.6 ± 4.1 nm (averaged across 60 randomly selected particles, Supplementary Fig. 1a) and single crystalline structure enclosed by {100} facet, in accordance with the atomic configuration diagram (Fig. 1c). The typical TEM and HRTEM images shown in Fig. 1d, e reveal that the synthesized Pd nanocrystals have concave structure of size about 41.2 ± 2.8 nm (Supplementary Fig. 1b), comparable to Pd NCs. Figure 1f shows the atomic configuration model of Pd CNCs enclosed by {730} facet. Figure 1g, h shows high-angle annular dark-field scanning TEM of Pd CNCs. The profile analogous to concave structure can be clearly observed and the angle between the {730} facets and the {100} facets is about 25.1 ± 3.6°, which is consistent with the model result shown in Fig. 1i. Overall, these results indicate that {100}-facet-enclosed Pd NCs and {730}-facet-enclosed Pd CNCs are successfully prepared.

**Oxidase-like activity of Pd nanocrystals**. The absorption spectra were measured at 370 and 652 nm, as 3,3,5,5-tetra-methylbenzidine (TMB) is oxidized. The characteristic peaks in Supplementary Fig. 2a show that TMB is oxidized by addition of the Pd nanocrystals. Moreover, the concave-structured Pd CNCs promote TMB oxidation more effectively than the Pd NCs (Supplementary Fig. 2a) and the oxidation rate is enhanced as the concentration of Pd CNCs increased (Supplementary Fig. 2b). Noble metal nanocrystals have been known to catalytically oxidize glucose, peroxidase substrates, and other organic compounds relevant to cellular respiration[23–27]. Here we measured the activation of $O_2$ by Pd nanocrystals, using TMB molecular probes. The concave-structured Pd CNCs maintain remarkable catalytic activity over a wide temperature range, indicating its excellent thermal stability (Supplementary Fig. 2c). The oxidation rate reaches a maximum at pH 4, then gradually decreases as pH value increases. The oxidation rate is decreased by about 80% from pH 4 to 8 (Supplementary Fig. 2d). These findings suggest that Pd nanocrystals, especially concave structures, could be excellent oxidation catalyst with high activities and stability.

**Oxidation of ascorbate catalyzed by Pd nanocrystals**. The oxidation of ascorbate was monitored by ultraviolet–visible (UV–vis) spectra. The absorption band is located at 265 nm. Figure 2a, b shows that the absorbance peaks are reduced when either Pd NCs or Pd CNCs are added, and the rate of the peak reduction is faster with Pd CNCs. Concave-structured Pd nanocrystals display higher catalytic activity on ascorbate oxidation than Pd NCs, following the same trend with the above data from the TMB oxidation measurement.

$O_2$ also participates as a reactant in ascorbate oxidation reaction. The consumption of $O_2$ has been monitored in either presence or absence of Pd nanocrystals, using electron spin resonance (ESR) oximetry. The reduction of $O_2$ concentration results in the progressive enhancement of the super hyperfine structure of 3-carbamoyl-2,5-dihydro-2,2,5,5-tetramethyl-1H-pyrrol-1-yloxyl (CTPO), which has been commonly used as the ESR spin label probe[28,29]. As displayed in Fig. 2c, the ESR spectrum of CTPO in the absence of Pd nanocrystals exhibit only a smooth profile over the sampled frequency range. The ESR

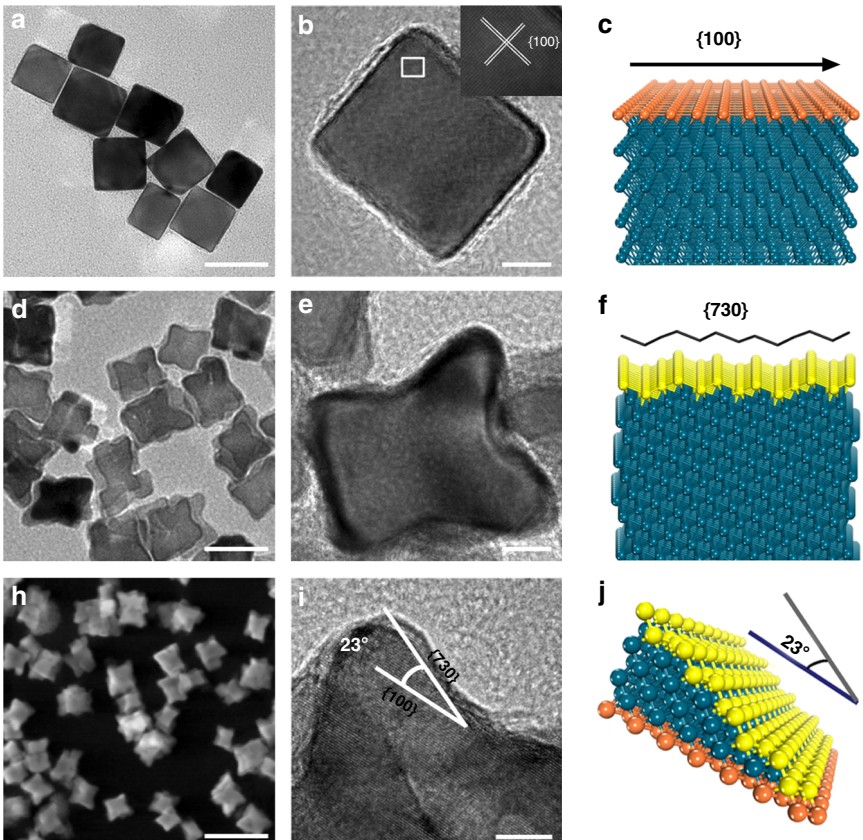

**Fig. 1** Characterization of the as-fabricated Pd nanocrystals. **a** TEM image of Pd nanocubes (Pd NCs) enclosed by {100} surface facets. Scale bar: 50 nm. **b** TEM image of a single Pd NC. Scale bar: 10 nm. **c** Atomic configuration diagram of Pd NCs. **d** TEM image of Pd concave nanocubes (Pd CNCs) enclosed by {730} surface facets. Scale bar: 50 nm. **e** TEM images of a single Pd CNC. Scale bar: 10 nm. **f** Atomic configuration diagram of Pd CNCs. **g** HAADF-STEM image of Pd CNCs. Scale bar: 100 nm. **h** High resolution TEM (HRTEM) image of Pd CNCs. Scale bar: 5 nm. **i** Model of the atomic arrangement on the {730} surface

spectra indicate that a sufficient amount of $O_2$ is dissolved in the solution, in line with the slow autoxidation of ascorbate in the absence of Pd nanocrystals. Meanwhile, in the presence of either Pd NCs or Pd CNCs, high intensity of super hyperfine structures appears, indicating that the consumption rate of $O_2$ is enhanced by Pd nanocrystals. Moreover, consistent with the above other measurements, the $O_2$ consumption rate increases more rapidly in the presence of Pd CNCs than Pd NCs, suggesting that concave-structured Pd nanocrystals are more effective in catalyzing the autoxidation of ascorbate.

The oxidation of ascorbate can be detected by the production of ascorbate radical, which is the intermediate of the oxidation reaction. Previous research has shown that ascorbate radical is generated in the early stage of ascorbate oxidation, then converted into DHA through one-electron oxidation reaction[30]. Ascorbate radical is detected by ESR spectroscopy. Figure 2d shows that extremely weak ESR signal was detected in the control sample (with ascorbate alone), indicating the negligible autoxidation of ascorbate itself under the current experimental conditions. In contrast, distinct ESR signals of ascorbate radicals were observed after adding Pd nanocrystals. The $H_2O_2$ formation during the oxidation of ascorbate was determined by using hydrogen peroxide assay kit in either the presence or the absence of Pd nanocrystals[31]. Production of $H_2O_2$ was measured in a time-dependent manner in the presence of Pd nanocrystals (Fig. 2e). The Pd CNCs significantly accelerate the $H_2O_2$ production over time, compared to the Pd NCs ($p < 0.01$).

To reveal the structural transformation of ascorbate in the presence of Pd CNCs, we analyzed its oxidation product using

mass spectrometry (MS). As shown in Supplementary Fig. 3, we found that (i) negative ion mode provide the best response for both ascorbic acid and DHA, where the deprotonated molecular ion $[M-H]^-$ at $m/z$ 172.90 and 175.00 can be monitored very well on the MS chromatograms, which can be attributed to the products DHA and substrate ascorbic acid, respectively; (ii) strong ascorbic acid and weak DHA signal was detected in the sample of ascorbic acid alone, indicating its slow rate of autoxidation under experimental conditions; and (iii) when Pd CNCs was added, DHA signal at $m/z$ 172.90 increased significantly, while ascorbic acid peak at $m/z$ 175.00 diminished. These results of MS analyses confirm that the oxidation reaction of ascorbate is catalyzed by Pd CNCs.

**Quantum mechanic calculations on the catalytic activities of Pd nanocrystals**. It was previously proposed that surface facet might be a key parameter to modulate the absorption and the activation of $O_2$ on metal nanocrystals[26,31]. Long et al.[32] has recently shown that electron transfer from Pd atom to the $O_2$ molecule is more facilitated when $O_2$ is absorbed by {100} surface than {111} surface. As electron density of the $O_2$ molecule is increased, its magnetic moment is reduced (2.0 μB for the isolated $O_2$, 0.017 μB for the $O_2$ adsorbed by Pd{100} surface, 0.549 μB for the $O_2$ absorbed by Pd{111} surface)[32].

Herein, we employed the first-principles density functional theory (DFT) to calculate the affinity of the absorption of $O_2$ on either Pd{730}, {100}, or {111} surface, where the first two types of the surfaces are taken from Pd CNCs and Pd NCs, respectively.

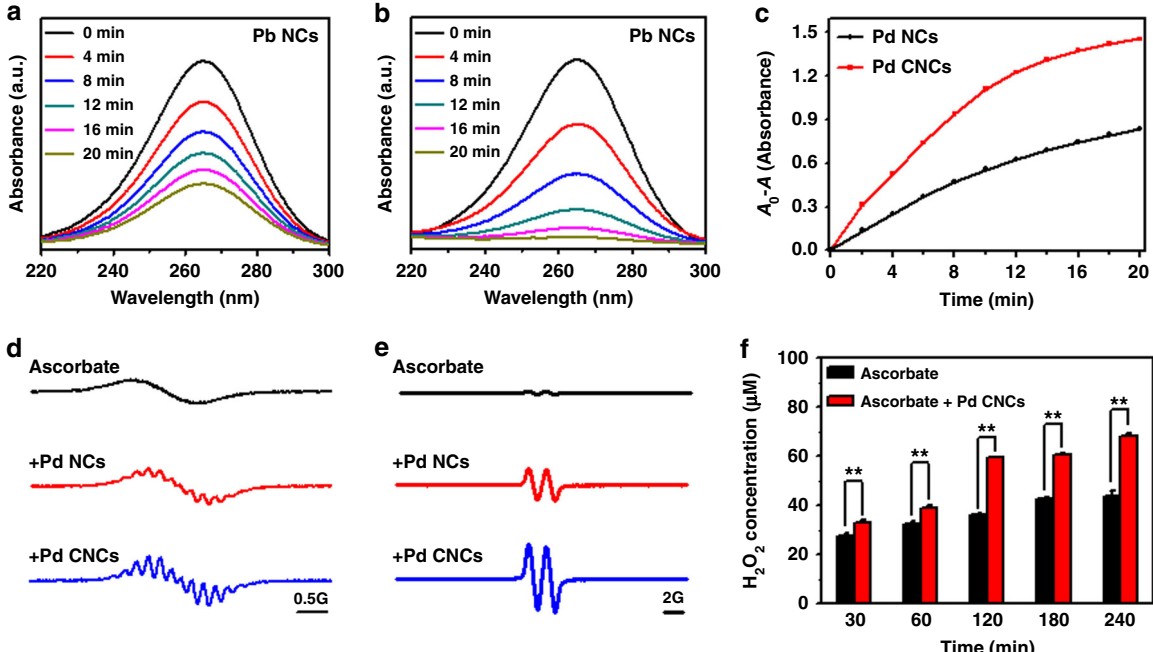

**Fig. 2** Oxidation of ascorbate catalyzed by Pd nanocrystals. **a, b** Evolution of UV-*vis* absorption spectra for 100 μM ascorbate with 50 μg/mL Pd nanocubes (Pd NCs) or Pd concave nanocubes (Pd CNCs) in aqueous solution. **c** Time-dependent absorbance changes of ascorbate at 265 nm with Pd NCs or Pd CNCs. **d** Oxygen consumption during oxidation of ascorbate. ESR spectra of 0.1 mM CTPO obtained from samples containing 5 mM ascorbate in the absence and presence of 50 μg/mL Pd nanocrystals at 10 min. **e** Ascorbic radical production. ESR spectra were collected at 2 min after mixing 5 mM ascorbate with or without 50 μg/mL Pd nanocrystals. **f** The concentration of $H_2O_2$ with ascorbate alone, or the combination of ascorbate and Pd CNCs. The numerical data are expressed as the mean ± standard deviation. Error bars were taken from three parallel experiments ($n = 3$). $P$ values compared to that of corresponding ascorbate alone treatment were calculated by Student´s $t$-test: $**p < 0.01$

The Pd{111} surface was also chosen for the DFT calculation as a control for the consistency in the setups with the previous work[32]. The DFT calculation results for {730} and {100} surfaces are shown in Fig. 3, and {111} in Supplementary Fig. 3. The {730} surface is an undulating surface composed of highly unsaturated 6-coordinated Pd atoms ($Pd^{6c}$). As shown in Fig. 3a, b, when $O_2$ molecule is absorbed onto {730} surface ("$O_2$@Pd{730}"), $O_2$ favors binding with two $Pd^{6c}$ atoms of one ridge on Pd{730} surface. The O–O distance was elongated to 1.351 Å (from the isolated $O_2$ bond length of 1.224 Å). The Pd NCs is a planar surface composed of 8-coordinated Pd atoms ($Pd^{8c}$). When $O_2$ is absorbed onto {100} surface ("$O_2$@Pd{100}"), the $O_2$ molecule lays between two lines of $Pd^{8c}$ atoms, forming four O–$Pd^{8c}$ bonds (Fig. 3d, e). The O–O distance of $O_2$@Pd{100} was 1.424 Å, (1.402 Å in ref. [32]), which is longer than that of $O_2$@Pd{730}. The longer O–O distance of $O_2$@Pd{100} is attributed to the stronger binding of each O with two Pd atoms on Pd{100} surface. However, the more unsaturated $Pd^{6c}$ atom of the Pd{730} surface lead to the shorter O–$Pd^{6c}$ distance (about 2.056 Å) than the O–$Pd^{8c}$ distance (about 2.152 Å). The Pd{111} surface is also a planar surface but composed of 9-coordinated Pd atoms ($Pd^{9c}$). For the $O_2$@ Pd{111} (see Supplementary Fig. 4), the $O_2$ molecule binds to only two $Pd^{9c}$ atoms and the calculated O–O and O–$Pd^{9c}$ distances are 1.353 Å (1.324 Å in ref. [32]) and 2.094 Å, respectively. In brief, the bond length of $O_2$ molecule was elongated in all three Pd surfaces, indicating $O_2$ molecule is activated.

As suggested in the previous study[32], electron transfer from the Pd surface to the $O_2$ molecule causes the activation of $O_2$ molecule. In this study, our calculated Hirshfeld charges on each O atom of $O_2$@Pd{730}, $O_2$@Pd{100}, and $O_2$@Pd{110} are $-0.208e$, $-0.205e$, and $-0.187e$, respectively. Although the coordination number of O–Pd pairs is smaller and the O–O

bond length is shorter in $O_2$@ Pd{730} system, the amount of the electron transferred from Pd to $O_2$ is similar to that in Pd{100}, and larger than Pd{111} surface. Moreover, the magnetic moment of $O_2$ molecule is decreased in both $O_2$@Pd{730} and $O_2$@Pd{100} systems. The calculated magnetic moments of $O_2$ in $O_2$@Pd{730}, $O_2$@Pd{100}, and $O_2$@Pd{111} are 0.000, 0.000 (0.017 μB in ref. [32]), and 0.562 μB (0.549 μB in ref. [32]), respectively. Our DFT calculation results indicate that the $O_2$ molecules on Pd{730} and Pd{100} surfaces are similarly activated.

We further calculated the electrostatic potential (ESP) distribution of the electron density on the van der Waals surface to quantitatively evaluate the electrostatic features of the $O_2$ molecules absorbed by either Pd{730} or Pd{100} surfaces. As shown in Fig. 3c, f, $O_2$ molecules on both Pd{730} and Pd{100} surfaces have negative values on the ESP. The minimum value of the ESP of $O_2$@Pd{730} ($-26.40$ kcal/mol) is lower than that of $O_2$@Pd{100} ($-21.94$ kcal/mol). $O_2$ molecule on Pd{730} surface, which carries more negative charge than that on Pd{100} surface, forms stronger hydrogen bond with the hydroxyl terminal of ascorbate. Oxygen on Pd{730} surface can serve more efficiently as a precursor for the subsequent oxidation reaction of ascorbate than that on Pd{100}.

We then further simulated the oxidation reaction of the ascorbate on both Pd{730} and Pd{100} surfaces. The key reaction step of the ascorbate oxidation is the H transfer from the hydroxyl terminal of the ascorbate to the oxidant. For example, when the $O_2$ molecule acts as the oxidant, the H transfer mechanism can be described as the reaction scheme as follows, ($2 \times -OH + O_2 \rightarrow 2 \times -O\cdot + H_2O_2$). We herein constructed a simple $CH_3OH$ model to mimic the hydroxyl terminal of the ascorbate in our calculations, and the reaction was assumed as ($CH_3OH + O_2$)@surface $\rightarrow$ ($CH_3O + O_2H$)@surface. For the reactant@Pd{730}, as shown in Fig. 4a, b, the hydroxyl group

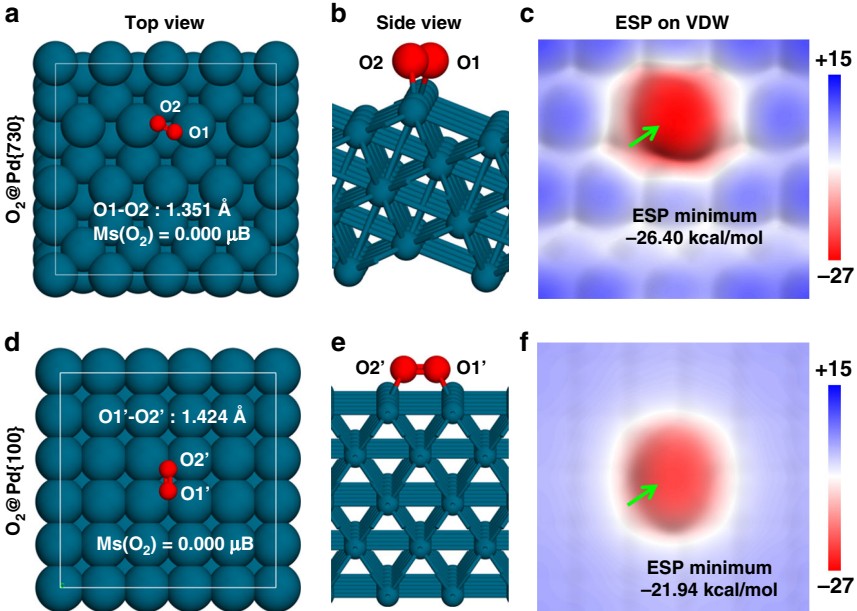

**Fig. 3** The dissociative adsorption of $O_2$ on the Pd {730} and Pd {100} facets. **a**, **b**, **d**, **e** Top and side views of DFT optimized structures of $O_2$ molecule adsorbing on Pd{730} and Pd{100} surfaces. **c**, **f** The electrostatic potential distributed on the electron density Van der Waals surfaces (isodensity=0.001 a.u.) of $O_2$@ Pd{730} and $O_2$@ Pd{100}. The green arrows in **c** and **f** point to the global EPS minima of the two electron density VDW surfaces

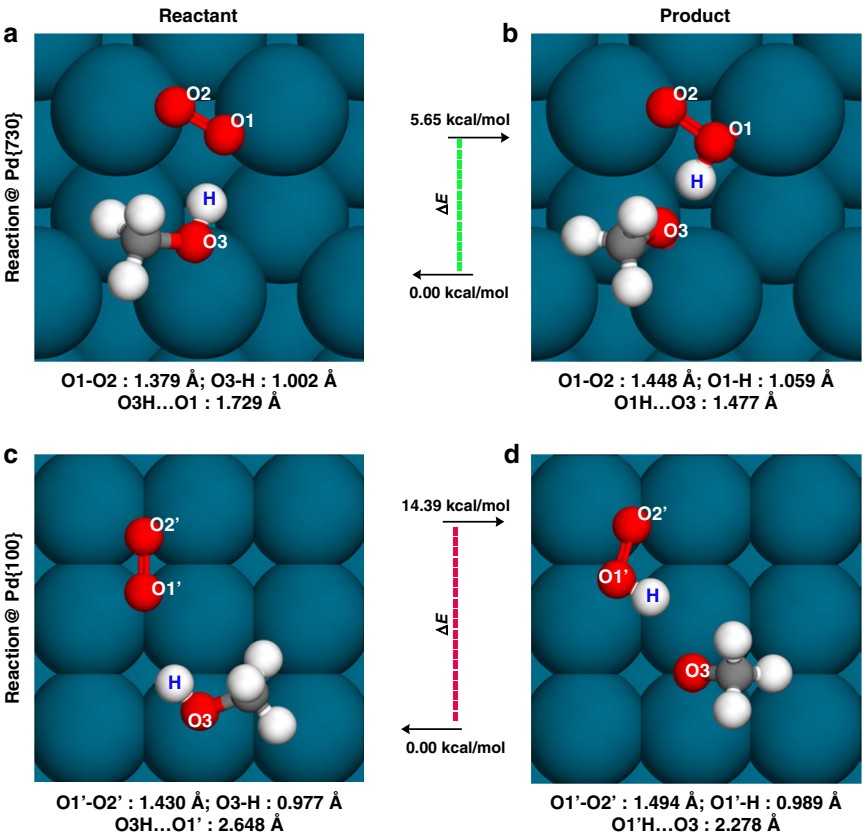

**Fig. 4** Simulated $CH_3OH$ oxidation on Pd{730} and Pd{100} surfaces. **a–d** represent the optimized structures of the reactants and the products on Pd {730} and Pd{100} surfaces. The $\Delta E$ values indicate the energy requirements of the endothermic reactions

also tended to be adsorbed by a ridge of Pd{730}, forming a very short hydrogen bond of 1.729 Å with $O_2$@Pd{730}. Hydrogen bond interaction between the hydroxyl group of $CH_3OH$ and $O_2$@Pd{730} is enhanced by a high density of negative charge on

$O_2$@Pd{730}. The $O_2$ bond length was further elongated to 1.379 Å. Meanwhile, the "O3-H" (see Fig. 4 for the nomenclature of "O1", "O2", and "O3" oxygen atoms) bond length of the hydroxyl was obviously elongated to 1.002 Å (compared to the same

distance of the isolated $CH_3OH$, 0.973 Å) caused by the strong "O3H···O1" hydrogen bond (Fig. 4a). The difference in the energy between reactant and product of the H transfer reaction, $(CH_3OH + O_2)@Pd\{730\} \rightarrow (CH_3O + O_2H\cdot)@Pd\{730\}$, is about 5.65 kcal/mol. Meanwhile, for the same reaction on Pd{100} surface, $(CH_3OH + O_2)@Pd\{100\}$, the hydroxyl group is bound to a bare $Pd^{8c}$ atom, close to the $O_2$ molecule. However, the overall effective negative charge density of $O_2@Pd\{100\}$ is smaller than that of $O_2@Pd\{730\}$, causing hydrogen bond interaction between $O_2$ and hydroxyl group weaker. Figure 4c, d shows the hydrogen bond distance is about 2.648 Å in the case of Pd{100}. The bond lengths of $O_2$ and the "O3-H" were only slightly elongated. In this case, the energy difference of the H transfer reaction on Pd{100} surface, $(CH_3OH + O_2)@Pd\{100\} \rightarrow (CH_3O\cdot + O_2H\cdot)@Pd\{100\}$, is about 14.39 kcal/mol, which is significant larger than the energy difference on Pd{730} surface. The much smaller energy requirement on the surface of concave-structured Pd nanocrystals for the hydrogen transfer from the hydroxyl terminal of ascorbate to $O_2$ molecule indicates a more reactive oxidation than that on Pd concave nanocubes. These DFT calculations of the oxidation reaction of the ascorbate provide a direct evidence and support for the above experimental observations of stronger catalytic activities of Pd CNCs.

**Pd nanocrystal-catalyzed ascorbate oxidation enhances its cytotoxicity to cancer cells.** In the current study, colorectal carcinoma cell lines HCT116 and intestinal epithelial cell lines IEC6 were chosen to test the cytotoxicity of ascorbate selectively against cancer cells. First, two types of cells were exposed to ascorbate at varied concentrations for 24 h. As shown in Supplementary Fig. 5a, the HCT116 cell lines treated with above 3 mM ascorbate show a remarkable decrease in the cell viability, and its 50% inhibitive concentration (IC50 value) is about 4.9 mM. In contrast, the IEC6 cell is insensitive to 5 mM ascorbate (see Supplementary Fig. 5b). These results indicate that ascorbate could selectively killed cancer cells HCT116 but not normal cells IEC6.

Next, the cytotoxic potential of ascorbate combined with Pd CNCs was examined. As indicated in Supplementary Fig. 6a, in the presence of 2.5 mM ascorbate, the Pd CNCs induce dose-dependent killing effects on tumor cells, while Pd CNCs alone do not show any toxic effect on the HCT116 cells (Supplementary Fig. 7a). However, under the same experimental conditions, no clear cell death is observed for normal cells (Supplementary Fig. 6b), indicating that Pd CNCs selectively enhanced the toxicity of ascorbate to cancer cells. In view of the Pd nanocrystals present no harmful effects (Supplementary Fig. 7b) on normal cells, 50 μg/mL Pd nanocrystals are employed in the following experiment.

As shown in Fig. 5a, the survival rate of cancer cells is not significantly decreased after addition of ascorbate up to 2 mM, in the absence of Pd CNCs. Meanwhile, upon addition of 50 μg/mL CNCs, the survival rate starts to decrease at lower ascorbate concentration. As the ascorbate concentration increased to 3 mM, the survival rate decreased by 20% with ascorbate alone. However, when Pd CNCs were added, the survival rate decreased by 85%. No obvious toxicity was observed in normal cells after being exposed to the same dose of ascorbate, no matter whether Pd CNCs were added (Fig. 5b). Live/dead assay was also conducted to verify the enhanced antineoplastic effects. The intensity of red fluorescence signals in Fig. 5c shows that the combined treatment of ascorbate and Pd CNCs induces significant increase in the number of dead cells. Taken together, all these results show that Pd CNCs could selectively amplify the cytotoxic effects of ascorbate toward cancer cells but not normal cells.

**Increased genotoxic and metabolic stresses from Pd-catalyzed ascorbate oxidation.** We then conducted a series of experiments to assess the response of cancer cells to the ascorbate-induced oxidative stress. $H_2O_2$ and DHA, generated by oxidized ascorbate in the plasma, diffuse into the cytosolic compartments and potentially induce oxidative stress in cells. We measure the level of reactive oxygen species (ROS) in HCT116 cells using 2',7'-dichlorodihydrofluorescein diacetate (DCFH-DA) fluorescence probes (Fig. 6a). The fluorescence signal in cells is extremely weak upon exposure to Pd nanocrystals, demonstrating that the peroxide level in cells is negligible. The signal becomes slightly stronger with ascorbate treatment, inducing a limited amount of peroxide. By contrast, a strong fluorescence signal is found in cells upon co-exposure to Pd nanocrystals and ascorbate, indicating the remarkable increase of the peroxide levels.

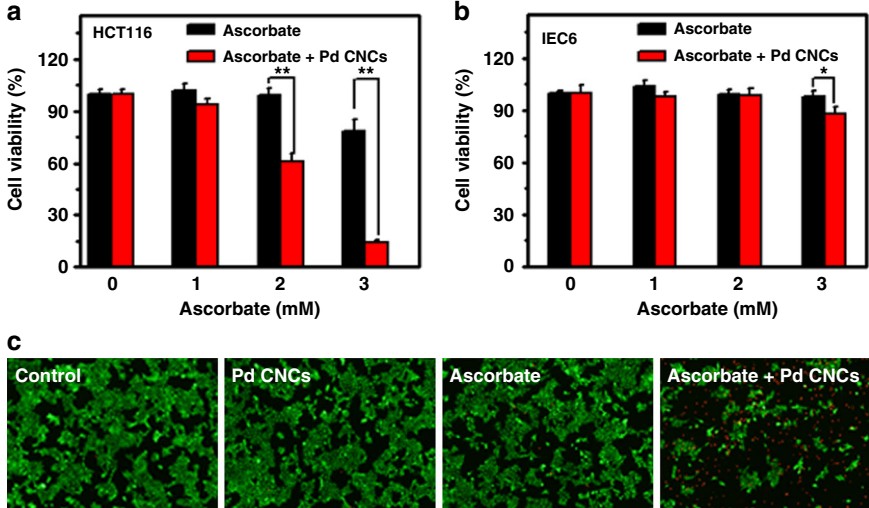

**Fig. 5** Effects of Pd CNC-catalyzed ascorbate oxidation on the cytotoxicity. Cell viability of HCT116 cells (**a**) and IEC6 cells (**b**) treated with either 1-3 mM ascorbate or 50 μg/mL Pd CNCs or combined ascorbate/ Pd CNCs for 24 h. Error bars were taken from three parallel experiments ($n = 3$). $P$ values compared to that of corresponding ascorbate alone treatment were calculated by Student´s $t$-test: $*p < 0.05$, $**p < 0.01$. **c** Fluorescent images of live and dead stains of HCT116 cells exposed to 2.5 mM ascorbate plus 50 μg/mL Pd CNCs. Scale bar = 100 μm

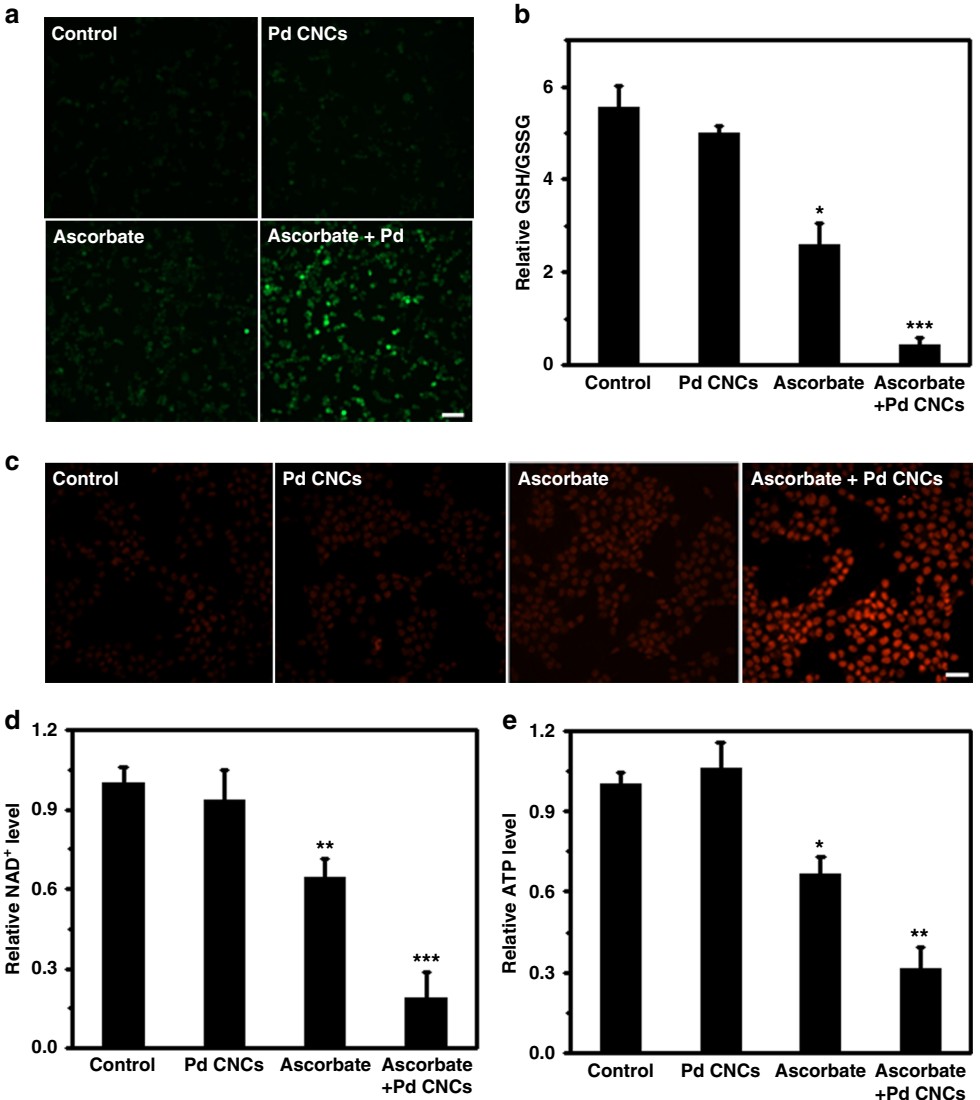

**Fig. 6** Mechanisms of ascorbate-induced cytotoxicity enhanced by Pd CNCs. **a** Representative fluorescence images of intracellular ROS levels stained with DCFH-DA probes. HCT116 cells were respectively treated with 2 mM ascorbate, 50 µg/mL Pd CNCs, or combined ascorbate/Pd CNCs. Scale bar: 70 µm. **b** Relative ratios of reduced to oxidized glutathione (GSH/GSSG) in HCT116 cells after various treatments. **c** Representative fluorescence images of fragmented DNA in HCT116 cells after various treatments using the TUNEL assay. Scale bar: 35 µm. **d** Changes in NAD$^+$ levels. **e** ATP levels in HCT116 cells after various treatments. Error bars were taken from three parallel experiments ($n = 3$). $P$ values compared to that of corresponding PBS-treated cells were calculated by Student´s $t$-test: *$p < 0.05$, **$p < 0.01$ and ***$p < 0.001$ vs control

It is expected that ROS overproduction will lead to the depletion of intracellular antioxidant systems. GSH is an important cellular antioxidant, and responsible for detoxifying excess $H_2O_2$, resulting in the formation of oxidized glutathione (GSSG). Meanwhile, the oxidized product, DHA, may also deplete cellular GSH as it was consumed in reducing DHA to ascorbate in cells. Therefore, the relative ratios of GSH to GSSG (GSH/GSSG) in cell lysates can be treated as an indicator of oxidative stress in cell lysates. The ratio of GSH/GSSG is slightly decreased upon exposure to Pd nanocrystals (Fig. 6b). Treatment with ascorbate results in a significant decrease in the ratio of GSH/GSSG ($p < 0.05$), whereas co-treatment with ascorbate and Pd CNCs extremely decreases the ratio of GSH/GSSG ($p < 0.001$). These findings show that exposure to ascorbate assisted with concave-structured Pd nanocrystals lead to much more remarkable oxidative stress in cells than ascorbate alone.

Excess ROS production causes oxidative DNA damage. We measured DNA damage using terminal deoxynucleotidyl transferase dUTP nick end-labeling assay. As shown in Fig. 6c,

low doses of ascorbate induce a mild DNA damage, while addition of Pd CNCs enhances the damage dramatically. DNA cleavage activates DNA damage repair enzyme and poly (ADP-ribose) polymerase (PARP) (Supplementary Fig. 8). Activated PARP leads directly to NAD$^+$ depletion followed by ATP depletion and cell death via necrosis or apoptosis. Consistently, the ascorbate combined with Pd nanocrystals significantly decreases the NAD$^+$ level than that of ascorbate-only treatment, indicating the potential ATP depletion. Intracellular NAD$^+$ level in HCT116 cells was remarkably decreased after combined treatment of ascorbate with Pd CNCs (Fig. 6d). ATP level in cells does not change much upon addition of Pd nanocrystals alone, however is significantly decreased by ascorbate treatment ($p < 0.05$), and further decreased by the combined treatment of ascorbate and Pd CNCs ($p < 0.01$). All these data show that concave-structured Pd nanocrystals potentiate the cytotoxic effects of ascorbate toward tumor cells, mainly via accelerated ascorbate oxidation-induced genotoxic stress and metabolic stress.

**Combined actions of Pd CNCs and ascorbate suppress tumor growth in vivo**. Next, we performed animal experiment to evaluate whether Pd nanocrystals could enhance the antitumor efficacy of ascorbate. Moreover, both oxaliplatin and 5-fluorouracil (5-FU), agents currently in clinical use for colon cancer, were selected as positive control to further evaluate the efficacy and safety of combined therapeutic regimen. Mice bearing human colorectal carcinoma HCT116 tumor xenografts were divided into six groups: group 1, treated with saline as control; group 2, with ascorbate; group 3, with Pd CNCs; group 4, combined ascorbate and Pd CNCs; group 5, with 5-FU; and group 6, with oxaliplatin. As seen in Fig. 7a, there were no significant changes in the weight of mice of all three treatment groups except for oxaliplatin, indicating both ascorbate and concave-structured Pd nanocrystals are well tolerated in vivo.

The tumor growth curves in Fig. 7b show that as compared to saline treated mice, treatment with Pd nanocrystals has no effect on the tumor growth, whereas ascorbate treatment display a limited inhibitory effect on the tumor growth. For the combined treatment group, with the presence of Pd CNCs, the ascorbate treatment remarkably suppresses tumor growth. Tumor volume is decreased by 53.9% when the combined ascorbate and Pd CNCs treatment is used, while it only decreased 27.5% for the ascorbate alone treatment (Fig. 7c). Moreover, in comparison with these currently in use first-line drugs, the antitumor effect of the Pd CNCs/ascorbate combination in HCT116 xenografts was similar to oxaliplatin, but more effective than 5-FU. In addition, administration of the combination did not result in any meaningful loss of body weight, while animals receiving the oxaliplatin regimen lost a significant amount of weight over the five cycles of agent administration. The time-dependent biodistribution experiment was carried out and the data show that these Pd CNCs were mainly accumulated in the reticuloendothelial system, such as liver and spleen (Supplementary Fig. 9). We further used metabolism cages to collect urine and feces of mice after injection of Pd CNCs, and a majority of the Pd CNCs were detected in the feces, suggesting efficient clearance through fecal excretion. All these data indicated that the combined treatment shows significant enhancement in the suppression of tumor growth in preclinical models.

## Discussion

Ascorbate has been developed as a potential anticancer agent in the last 50 years, and typically treated with other chemotherapeutic agents, in order to enhance its antitumor activity. In this study, we proposed to apply Pd nanocrystals as a new additive agent to catalyze the oxidation of ascorbate and thus to enhance the ascorbate-induced oxidative stress to cancer cells. We further demonstrated that we can tune the surface structure of Pd nanocrystals through rational design to achieve better efficacy. The concave-structured Pd NCs enclosed by high-index facets were found to exhibit higher catalytic activity than Pd NCs enclosed by low-index facets—i.e., the former can more efficiently accelerate the autoxidation of ascorbate and generate more $H_2O_2$ than the later. Moreover, our first-principles calculations revealed the underlying reaction mechanisms for the activation of $O_2$ and the oxidation of ascorbate. When $O_2$ molecule is chemically absorbed by the surface of concave-structured Pd NCs, electron transfer from Pd atom to $O_2$ molecule becomes more favorable. As $O_2$ molecule becomes more electronegative, it forms a stronger hydrogen bond with the hydroxyl terminal of the incoming ascorbate. The energy requirement of the hydrogen transfer from the hydroxyl terminal of the ascorbate to the $O_2$ molecule on the surface of concave-structured Pd NCs is smaller than that on Pd NCs.

Our cell viability experiments showed that concave-structured Pd NCs remarkably amplifies the cytotoxic effect of ascorbate, which is selective against cancer cells. The viability of normal cells was not affected by addition of Pd nanocrystals. The enhancement of the cytotoxic effect of ascorbate by Pd nanocrystals was observed in other cellular responses to the oxidative stresses, such as elevated $H_2O_2$-mediated genotoxic (DNA damage) and metabolic (ATP depletion) stresses. Finally, the in vivo animal experiment confirmed that the combined treatment with ascorbate and Pd nanocrystals suppresses tumor growth more efficiently than ascorbate alone.

In summary, we performed various in vitro and in vivo experiments to demonstrate a potential use of Pd nanocrystals to enhance the anticancer efficacy of ascorbate, and our quantum mechanic calculations further reveal the underlying mechanism for their catalytic activity. We anticipate that the application of Pd nanocrystals might show promise in the development of novel ascorbate-based antitumor agents.

## Methods

**Chemicals**. L-ascorbic acid (molecular weight (M.W.) = 176.12 Da), ascorbate (M.W. = 198.11 Da), potassium bromide (KBr, M.W. = 119.00 Da), poly(vinyl pyrrolidone) (PVP, M.W. = 55,000 Da), ruthenium chloride hydrate (RuCl₃, M.W. = 207.43 Da), sodium palladium (II) chloride (Na₂PdCl₄, M.W. = 294.21 Da), TMB (M.W. = 240.34 Da), and CTPO (M.W. = 183.23 Da) were

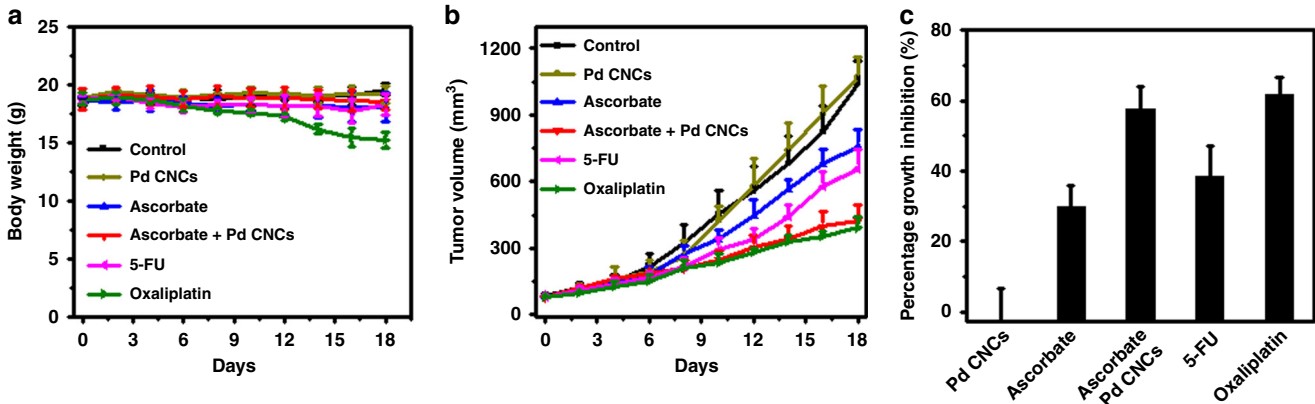

**Fig. 7** Effects of Pd CNCs on ascorbate-induced cytotoxicity in vivo. The current clinical medicines, oxaliplatin and 5-fluorouracil (5-FU), are also included for comparison. **a** Average weight of mice with HCT116 xenografts after different treatments. **b** Tumor growth curves from HCT116 tumor xenografts in 6–8 week athymic nude mice with ascorbate or Pd CNCs. **c** Percentage of growth inhibition of mice after the treatment. For all in vivo studies, mice n = 5 per treatment group

purchased from Sigma Aldrich and used as received. Ultrapure water (18.2 MΩ, Millpore Co.) was used throughout each experiment.

**Synthesis and characterization of the Pd nanostructures**. The direct synthesis of concave-structured Pd nanocrystals with exposed high-index {730} facets were carried out using a previously reported protocol[26]. Briefly, 3 mL of aqueous $Na_2PdCl_4$ solution (57 mg) was introduced into an aqueous solution (16 mL) containing PVP (210 mg), ascorbic acid (120 mg), and KBr (600 mg), which had been stirred at 80 °C for 30 min. Subsequently, 1 mL aqueous solution of $RuCl_3$ (5 mg) was injected and the reaction was maintained at 80 °C for another 30 min. Afterward, The Pd nanocrystals were separated by centrifugation, and washed with ultrapure water. The Pd NCs used as reference was prepared by a seed-mediated approach as described earlier[26,33]. A FEI Tecnai F20 TEM instrument was used to characterize the morphology and the size of as-prepared Pd nanocrystals.

**Oxidase-like activity of Pd nanocrystals**. A unit of 0.8 mM TMB was dissolved in phosphate-buffered saline (PBS) solution (pH 7.4) and 10 μL of aqueous suspension of different Pd nanocrystals (final concentration is 5 μg/mL) was then introduced at room temperature for 30 min. The samples were taken for UV–vis measurements at the wavelength range from 200 to 1000 nm using Shimadzu UV-3600 UV–vis spectrophotometer. Kinetic measurements of the oxidase reactions of concave-structured Pd nanocrystals (0–20 μg/mL) were performed by monitoring TMB absorbance at 652 nm.

**Oxidation of ascorbate catalyzed by Pd nanocrystals**. UV–vis spectroscopy of ascorbate: The oxidation of 100 μM ascorbate, in the presence or absence of 50 μg/mL Pd nanocrystals, was recorded by UV–vis measurement for 20 min in the wavelength range.

The consumption of $O_2$: ESR spin label oximetry was used to detect oxygen consumption during the reaction. Samples containing 5 mM ascorbate, 0.1 mM CTPO, and 2 μg/mL Pd nanocrystals were pipetted into capillary tubes. The ESR signals were recorded using a Bruker EMX ESR spectroscopy with 1 mW microwave power, 0.05 G modulation amplitude, and 5 G scan range. The ESR spectra were recorded at 5 min.

The generation of ascorbate radicals: The ESR spectrum of ascorbate radical (about 10 min half-life) was recorded at 5 min after 5 mM ascorbate is mixed with or without 2 μg/mL Pd nanocrystals. The measurements were carried out under the following conditions: microwave power 20 mW, field modulation 1 G, and scan width of 25 G.

The production of $H_2O_2$: A $H_2O_2$ assay kit (Beyotime Institute of Biotechnology, Shanghai, China) was used to identify the other product of reaction, $H_2O_2$. In brief, after mixing ascorbate with Pd nanocrystals for different time, the mixture was centrifuged (14,800 rpm, 10 min) and the supernatant solution was collected. The detection solution then was added to the supernatants for 30 min at room temperature and measured using a BioTek Synergy NEO microplate reader.

**DFT calculations**. Spin-polarized DFT calculations were performed using the Dmol3 program[34,35]. The Pd{730}, Pd{100}, and Pd{111} surfaces were modeled with 4 × 1 square slab with 12 layers (15.95 Å × 15.18 Å, 96 Pd atoms), 5 × 5 square slab with 5 layers (14.10 Å × 14.10 Å, 125 Pd atoms), and 5 × 5 rhombic slab with 5 layers (14.10 Å × 14.10 Å, 125 Pd atoms), respectively, and the vacuum region was set to 30 Å. Then, $O_2$ molecules, as well as the $CH_3OH$ molecules, were placed onto these slab to explore the adsorptions and the H transfer reactions. During geometry optimizations, the lattice and the Pd atoms were freezed and other atoms were allowed to relax. The Perdew–Burke–Ernzerhof (PBE) exchange–correlation function[36] with the Grimme scheme dispersion correction[37] was employed. The double-numerical basis set with polarization functions[35] was applied for all atoms. For Pd atoms, the DFT-semi core pseudopots approach[38] was used as the core treatment to reduce the computational consume. This approach can also introduce some degree of relativistic correction into the core. The conductor-like screening model[39,40] was used to simulate a solvent environment. The first Brillouin zone was sampled with 2 × 2 × 1 k-points and the global cutoff was set to 4.5 Å. Based on the optimized geometry structures, high precision energy calculations were performed using 7 × 7 × 1 k-points. For the optimizations of isolated $O_2$ and $CH_3OH$ molecules, they were placed into 50 Å × 50 Å × 50 Å empty boxes. Both geometry optimizations and energy calculations of these isolated molecules were performed using gamma k-points. A small smearing value of 0.005 a.u. was used in all calculations.

**Cell culture and viability assays**. The human colorectal carcinoma HCT116 cells (American Type Culture Collection (ATCC) Number: CCL-247) and the rat intestinal epithelial IEC6 cells (ATCC Number: CRL-1592) were obtained from the Cell Bank of the Chinese Academy of Sciences (Shanghai, China). Cells were cultured in high-glucose Dulbecco's modifed Eagle medium (HyClone), supplemented with 10% fetal bovine serum (FBS, GibcoBRL) and 1% penicillin/streptomycin (HyClone) at 37 °C with 5% $CO_2$ in a humidified incubator. A CCK-8 assay kit (Dojindo Laboratories) was used to detect the cell viability. Briefly, HCT116 cells or IEC6 cells were seeded on 96-well plates at a density of $5 \times 10^3$ cells/well, and cultured in complete culture medium containing 10% FBS. After

24 h, the medium was replaced by either ascorbate (0–10 mM) alone, Pd CNCs (0–100 μg/mL) alone, or the combination of both agents. The stock solution of ascorbate was made fresh before each experiment. The cells were washed with PBS after 24 h of incubation, and CCK-8 reagent was added to each well 1 h before measuring the optical density by a microplate reader at 450 nm.

The cell viability after different treatments was further determined using a live/dead fluorescent staining kit (Sigma, USA), according to the manufacturer's protocol. After treatments, cells were dispersed in serum-free buffer containing ethidium homodimer-1 and Calcein AM. The labeled cells were analyzed by Olympus fluorescence microscopy after 30 min of incubation.

**Intracellular ROS measurement**. The DCFH-DA dye was employed to determine the intracellular ROS level. HCT116 cells were seeded in confocal microscope dishes at a density of $2 \times 10^5$ cells/well and respectively treated with ascorbate or Pd CNCs as described above. Then, cells were labeled with DCFH−DA for 30 min in the dark. After being washed with PBS, intracellular ROS levels were measured using an Olympus FV1200 confocal laser microscope at 488 nm excitation wavelength.

**Western blots**. After treatment as described above, protein was extracted using IP lysing buffer, and the level of protein was determined by a BCA Protein Assay Kit. Fifty micrograms of protein lysates were separated on SDS/polyacrylamide gel electrophoresis gel, and the separated proteins were transferred onto polyvinylidene fluoride membrane, then nonspecific binding was blocked using 5% nonfat dry milk in PBS-Tween (0.2%) for 1 h. Afterward, the membranes were incubated with primary PARP antibody (cat. no. ab137653; Abcam), which were diluted 1:1000 in Tris-buffered saline containing 5% nonfat milk. Following PBS-Tween washes (three times), the membranes were blotted with secondary antibodies (1:25,000; cat. no. ab205718; Abcam) that were conjugated with horseradish peroxidase for 1 h.

**Xenograft and treatment procedures**. All animal experiments were conducted under protocols approved by the Soochow University Laboratory Animal Center. HCT116 cells ($2 \times 10^6$) suspended in PBS were injected subcutaneously into the flank of female athymic nude mice (6–8 weeks old). When tumor volume reached approximately 80 $mm^3$, mice were randomly divided into six groups (five mice/group), including "controls" that received 200 μL PBS; "Pd CNCs" 1 mg/kg intraperitoneal (i.p.); "ascorbate" 4 g/kg; "ascorbate plus Pd CNCs" (dosing and administration was the same as above); "5-FU" 10 mg/kg; and "oxaliplatin" that received 5 mg/kg. All the mice were treated with i.p. injection twice a day. Average weight and tumor size were measured every other day with vernier caliper, and tumor volume was calculated according to the following formula: tumor volume = $(L \times W^2) \times 0.5$, where $L$ is length and $W$ is width. Animals were euthanized and sacrificed when the tumor length exceeded 1.5 cm in any dimension. Statistics are based on standard deviations of six mice/group.

**Statistical analysis**. All the numerical data are expressed as the mean ± standard deviation. Statistical analysis was carried out on the values obtained from at least three independent experiments using two-tailed heteroscedastic Student's t-tests.

## Data availability
The authors declare that the data supporting the findings of this study are available within the paper and the Supplementary Information, or are available from the authors upon request.

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

## Acknowledgements

This work was partially supported by the National Basic Research Program of China (973 Program Grant No. 2014CB931900), National Natural Science Foundation of China (11575123, 11574224, and 21320102003), Collaborative Innovation Center of Radiological Medicine of Jiangsu Higher Education Institutions, and Jiangsu Provincial Key Laboratory of Radiation Medicine and Protection. Y.C. appreciates the support from the Natural Science Foundation of Jiangsu Province (BK20170353).

## Author contributions

The study was planned and directed by C.C.G. and R.H.Z. The nanomaterials were prepared by G.F. Experiments were conducted by Y.C. R.F.W., L.Z., X.T. and X.C.M. contributed to the improvement of the characterization of the materials. Quantum chemical calculation was conducted by X.D. and C.Z. The manuscript was prepared by C.C.G. and R.H.Z. S.L., C.Y.C. and Z.F.C. contributed to the improvement of the manuscript.

## Additional information

**Competing interests:** The authors declare no competing interests.

