## [Peer Review File · Nature Communications]

Reviewer #1 (Remarks to the Author):

Major claims of the paper: The paper claims that palladium nanocrystals can be used to oxidize ascorbate, a reaction that can be used in cancer therapy. A number of palladium nanoparticle morphologies are designed and tested and it is claimed that the reaction is specific to the shape of the nanocrystal. Overall this is a well written paper and the data shown is well organized.

Despite the good presentation, the results are rather surprising. Pd NPs are well known to produce ROS, and so are many other metallic NPs. So the overall result of having Pd in combination with the ascorbate result in oxidation is expected. What is surprising is that the ROS generated does not affect healthy/normal cells. This is an interesting result though it is not clear how well this is relative to other cells (perhaps not tested in this paper). This is a significant claim and I would like to suggest that other cells are tested.

The other issue is trying to determine the molecular mechanism for the reaction of ascorbate oxidation. Products have not been analyzed using molecular structural techniques such as nuclear magnetic resonance spectroscopy or mass spectrometry. These are important techniques that could shed significant information on the molecular mechanisms by which ascorbate is oxidized.

I believe the article will inspire thinking in the field, but before that it is important to get a better experimental understanding of the molecular mechanisms involved given the major claims made in this article.

Overall this is interesting work.

Reviewer #2 (Remarks to the Author):

The manuscript deals with the application of concave-structured palladium nanocrystals for the treatment of cancer, in particular colon rectal cancer, synergistically with ascorbate. The authors demonstrate via *in silico*, *in vitro* and *in vivo* characterizations that concave-structured nanocrystals of palladium can boost the oxidation of ascorbate, enhancing the production of oxidative stresses specifically on cancer cells.

The manuscript is clearly written. However, before suggesting its publication of Nature Communications, the following comments should be addressed:

1. the authors should more accurately characterize the differences between the Pd nanocubes and concave-structured Pd nanocrystals. In particular, the average size and related standard deviation should be provided for both crystals. Similarly the characteristic angle of the concave-structured crystals should be documented on a sufficiently large number of crystals (generally > 50). Average and standard deviation values should be included;
2. labels in Figure.2, 6 and 7 are too small and difficult to read;
3. for the tumor response, the authors should provide a direct comparison with a clinical standard approach for colon rectal cancer (ie. oxaliplatin or other similar chemotherapeutic drugs currently used for the successful treatment of colon rectal cancer);
4. the authors should comment on the fact that palladium is not a biodegradable material and 40 nm crystals would not be easily excreted through the kidneys. As such, the presented approach is relevant if and only if it is demonstrated to be superior to the current clinical standards;
4. the initial size of the tumor is extremely small (about 50 mm³). This should more considered as a preventive rather than a curative study.

Re: Manuscript ID: NCOMMS-18-05843

Dear Dr. Ylenia Lombardo,

We sincerely thank you for your prompt response to our submission and also appreciate both referees' very positive and constructive comments, which have further improved our manuscript.

We have identified the oxidized products of ascorbate in the presence of Pd concave nanocubes by mass spectrometry (Figure S2) as suggested by Referee #1; and carried new *in vivo* experiments which compares our antineoplastic protocols with current clinical first-line medicines for colon rectal cancer (Figure 7) as requested by Referee #2. All these new data further support our conclusions. We have also addressed all other minor points and revised the manuscript accordingly (with changes colored in blue). A point-by-point response to the referees' comments are listed on the following pages.

Finally, we would like to express our gratitude to you for overseeing this review process. We hope the revised manuscript is now acceptable for publication in Nature Communications.

Look forward to hearing from you soon.

Sincerely,

Ruhong Zhou, PhD

IBM Research and Columbia University

Reviewer #1 (Remarks to the Author):

Major claims of the paper: The paper claims that palladium nanocrystals can be used to oxidize ascorbate, a reaction that can be used in cancer therapy. A number of palladium nanoparticle morphologies are designed and tested and it is claimed that the reaction is specific to the shape of the nanocrystal. Overall this is a well written paper and the data shown is well organized.

RESPONSE: Thanks for the very positive comment. We are grateful for the time and effort expended by the reviewer.

1. Despite the good presentation, the results are rather surprising. Pd NPs are well known to produce ROS, and so are many other metallic NPs. So the overall result of having Pd in combination with the ascorbate result in oxidation is expected. What is surprising is that the ROS generated does not affect healthy/normal cells. This is an interesting result though it is not clear how well this is relative to other cells (perhaps not tested in this paper). This is a significant claim and I would like to suggest that other cells are tested.

RESPONSE: We thank the reviewer for this concern and giving us the opportunity to explain this result. In our study, the as-prepared Pd NPs showed no obvious cytotoxicity in concentrations up to 100 $\mu\text{g/mL}$ for both cancer and normal cells, as shown in Figure S6. This good biocompatibility may come from the presence of remnants of the PVP stabilizer on the surface of the Pd NPs.

However, in the presence of ascorbate, Pd NPs could selectively kill the tumor cells. That is because the differential sensitivity between cancer cells and normal cells to the product of ascorbate oxidation, H_2O_2 . Altered redox-active iron metabolism in cancer cells makes them more sensitive to the change of H_2O_2 level than normal cells. ^[1]

In addition, in order to further confirm this selective killing effect, other cells, including murine CT26 colon carcinoma cells, and human NCM-460 colon mucosal epithelial cells, are tested. Similarly, the survival rate of tumor CT26 cells decreased significantly in the presence of ascorbate combined with Pd NPs, while no obvious

toxicity were observed in the normal NCM-460 cells under the same experiment conditions (Figure R1).

Figure R1 Effects of Pd concave nanocubes (Pd CNCs) catalyzed ascorbate oxidation on the cytotoxicity for CT26 and NCM-460 cells. Cell viability of CT26 cells (a) and NCM-460 cells (b) treated with either ascorbate (1-3 mM) or 50 $\mu\text{g}/\text{mL}$ Pd CNCs or combined ascorbate/ Pd CNCs for 24 h. *P* values compared to that of corresponding ascorbate alone treated cells were calculated by Student's *t* test: **p* < 0.05, ***p* < 0.01, n=3.

[1] Doskey C M, Buranasudja V, Wagner B A, et al. Tumor cells have decreased ability to metabolize H_2O_2 : Implications for pharmacological ascorbate in cancer therapy. *Redox biology*, 2016, 10: 274-284.

2. The other issue is trying to determine the molecular mechanism for the reaction of ascorbate oxidation. Products have not been analyzed using molecular structural techniques such as nuclear magnetic resonance spectroscopy or mass spectrometry. These are important techniques that could shed significant information on the molecular mechanisms by which ascorbate is oxidized.

RESPONSE: We thank the reviewer for this valuable suggestion. To confirm the structural transformation of ascorbate in the presence of Pd concave nanocubes, we analyzed its oxidation product using mass spectrometry (MS). As shown in Figure S2, we found that (i) negative ion mode provide the best response for both ascorbic acid

and DHA, where the deprotonated molecular ion $[M-H]^-$ at m/z 172.90 and 175.00 can be monitored very well on the MS chromatograms, which can be attributed to the products DHA and substrate ascorbic acid, respectively; (ii) strong ascorbic acid and weak DHA signal was detected in the sample of ascorbic acid alone, indicating its slow rate of autoxidation under experimental conditions; and (iii) when Pd concave nanocubes was added, DHA signal at m/z 172.90 increased significantly, while ascorbic acid peak at m/z 175.00 diminished. These results of MS analyses confirm that the oxidation reaction of ascorbate is catalyzed by Pd concave nanocubes.

3. I believe the article will inspire thinking in the field, but before that it is important to get a better experimental understanding of the molecular mechanisms involved given the major claims made in this article.

Overall this is interesting work.

RESPONSE: We are greatly encouraged by the very positive comments of the Reviewer, and would like to express our gratitude again for your appreciation of our results, as well as the critical reading and helpful suggestions.

Reviewer #2 (Remarks to the Author):

The manuscript deals with the application of concave-structured palladium nanocrystals for the treatment of cancer, in particular colon rectal cancer, synergistically with ascorbate. The authors demonstrate via *in silico*, *in vitro* and *in vivo* characterizations that concave-structured nanocrystals of palladium can boost the oxidation of ascorbate, enhancing the production of oxidative stresses specifically on cancer cells.

The manuscript is clearly written. However, before suggesting its publication of Nature Communications, the following comments should be addressed:

RESPONSE: Thank you for your positive and constructive comments, which have improved our manuscript (details below).

1. The authors should more accurately characterize the differences between the Pd

nanocubes and concave-structured Pd nanocrystals. In particular, the average size and related standard deviation should be provided for both crystals. Similarly the characteristic angle of the concave-structured crystals should be documented on a sufficiently large number of crystals (generally > 50). Average and standard deviation values should be included;

RESPONSE: Thanks for the reviewer's suggestion. We have added the statistics of size and the characteristic angle of as-synthesized Pd nanocrystals, measured by TEM and HRTEM. As shown in Figure 1, these Pd concave nanocubes exhibited relatively uniform size of approximately 39.6 ± 4.1 nm in diameter, averaged across 60 randomly selected particles, which is comparable to synthesized Pd nanocubes (41.2 ± 2.8 nm). Moreover, the angle of between the {730} facets and the {100} facets is about $25.1 \pm 3.6^\circ$, measured from the 60 randomly selected particles.

2. Labels in Figure.2, 6 and 7 are too small and difficult to read;

RESPONSE: Thank you for pointing out this problem. They are all fixed -- we have removed the fuzzy picture labels and replaced them with a set of higher resolution images.

3. For the tumor response, the authors should provide a direct comparison with a clinical standard approach for colon rectal cancer (ie. oxaliplatin or other similar chemotherapeutic drugs currently used for the successful treatment of colon rectal cancer);

RESPONSE: Thanks for this valuable comment. As suggested, both oxaliplatin and 5-fluorouracil (5-FU), agents currently in clinical use for colon cancer, were selected as positive control to further evaluate the efficacy and safety of combined therapeutic regimen. In comparison with these first-line drugs, the anti-tumor effect of the Pd concave nanocrystals/ascorbate (Pd/Vc) combination in HCT116 xenografts was comparable to oxaliplatin but more effective than 5-FU (Figure 7). In addition, administration of the Pd/Vc combination did not result in any meaningful loss of body weight, while animals receiving the oxaliplatin regimen lost a significant amount of

weight over the five cycles of agent administration. All these data indicated that the application of Pd nanocrystals might indeed show promise in the development of novel ascorbate-based antitumor agents.

4. The authors should comment on the fact that palladium is not a biodegradable material and 40 nm crystals would not be easily excreted through the kidneys. As such, the presented approach is relevant if and only if it is demonstrated to be superior to the current clinical standards.

RESPONSE: We thank the Reviewer for this concern. We have carried out time-dependent bio-distribution experiment to investigate the clearance behaviors of Pd nanoparticles. The results showed that these Pd NPs could be efficiently cleared from the mouse body, with a majority of the Pd NPs eliminated through fecal excretion. In the future work, we will fine tune the size and surface property of nanoparticles to increase the renal excretion of nanoparticles.

5. The initial size of the tumor is extremely small (about 50 mm³). This should more considered as a preventive rather than a curative study.

RESPONSE: Thank you for this good point. The initial tumor volume of mice have been further increased to approximately 80 mm³ in our renewed *in vivo* experiments in comparison with oxaliplatin and 5-fluorouracil (5-FU), as mentioned above, which is widely used in mouse model for optimal cancer therapy. As shown in the new Figure 7, the combined treatment of Pd/Vc remarkably suppresses tumor growth, as effective as the current clinical first-line medicines, but with less side effects (weight loss). Therefore, we believe the application of Pd nanocrystals might indeed show promise in the development of novel ascorbate-based antitumor agents.

Reviewer #2 (Remarks to the Author):

The authors have significantly revised the manuscript, including additional animal experiments and a direct comparison with current clinical standards for CRC treatment.

However, before publication in Nature Communications, this reviewer strongly encourage the authors to more clearly address the following comments:

1. as per the nanocrystal size, the authors should provide the actual size distribution for the 60 tested nanocrystals as a bar chart documenting size (x-axis) and occurrence (y-axis);
2. as per the biodegradation or excretion of Pd nanocrystals, the authors claim that biodistribution studies have been performed and that "Pd NPs could be efficiently cleared from the mouse body, with a majority of the Pd NPs eliminated through fecal excretion." There is no data supporting this statement, neither in the main manuscript nor in the Supplementary Material. This reviewer would encourage the authors to include their data on biodistribution and fecal excretion in the main manuscript.
3. as per the initial tumor size, 80 mm³ is still insufficiently large for an anticancer treatment. Proper studies on tumor regression should consider an initial tumor size in the order of 200 - 300 mm³ (at least 10-15 days post cell inoculation) and follow the tumor growth up to 1,000 mm³, for all experimental groups. This would provide a more clear indication of the advantage of this therapy over conventional approaches.

REVIEWERS' COMMENTS:

Reviewer #2 (Remarks to the Author):

The authors have significantly revised the manuscript, including additional animal experiments and a direct comparison with current clinical standards for CRC treatment. However, before publication in Nature Communications, this reviewer strongly encourage the authors to more clearly address the following comments:

RESPONSE: Thanks for the very positive comment. We are grateful for the time and energy expended by the reviewer.

1. as per the nanocrystal size, the authors should provide the actual size distribution for the 60 tested nanocrystals as a bar chart documenting size (x-axis) and occurrence (y-axis);

RESPONSE: We thank the reviewer for this suggestion and the statistics of size distribution have been provided as Supplementary Figure 1.

2. As per the biodegradation or excretion of Pd nanocrystals, the authors claim that biodistribution studies have been performed and that "Pd NPs could be efficiently cleared from the mouse body, with a majority of the Pd NPs eliminated through fecal excretion." There is no data supporting this statement, neither in the main manuscript nor in the Supplementary Material. This reviewer would encourage the authors to include their data on biodistribution and fecal excretion in the main manuscript.

RESPONSE: As per the Reviewer's suggestion, we have provided the data on biodistribution and excretion of Pd concave nanocubes as Supplementary Figure 9.

3. As per the initial tumor size, 80 mm³ is still insufficiently large for an anticancer treatment. Proper studies on tumor regression should consider an initial tumor size in the order of 200 - 300 mm³ (at least 10-15 days post cell inoculation) and follow the tumor growth up to 1,000 mm³, for all experimental groups. This would provide a more clear indication of the advantage of this therapy over conventional approaches.

RESPONSE: We thank the Reviewer again for a very relevant comment. In our study, HCT116 cells were injected subcutaneously into the flank of female athymic nude mice. When the initial tumor size reaches 200 - 300 mm³, the tumor growth is so fast that the treatment time is not enough. In the future work, we will try to use other cell lines or tumor model to enlarge initial tumor size.